# Autoregressive Image Generation without Vector Quantization

**Tianhong Li**[1] **Yonglong Tian**[2] **He Li**[3] **Mingyang Deng**[1] **Kaiming He**[1]

[1]MIT CSAIL     [2]Google DeepMind     [3]Tsinghua University

## Abstract

Conventional wisdom holds that autoregressive models for image generation are typically accompanied by vector-quantized tokens. We observe that while a discrete-valued space can facilitate representing a categorical distribution, it is not a necessity for autoregressive modeling. In this work, we propose to model the *per-token* probability distribution using a diffusion procedure, which allows us to apply autoregressive models in a continuous-valued space. Rather than using categorical cross-entropy loss, we define a *Diffusion Loss* function to model the per-token probability. This approach eliminates the need for discrete-valued tokenizers. We evaluate its effectiveness across a wide range of cases, including standard autoregressive models and generalized *masked autoregressive* (MAR) variants. By removing vector quantization, our image generator achieves strong results while enjoying the speed advantage of sequence modeling. We hope this work will motivate the use of autoregressive generation in other continuous-valued domains and applications. Code is available at https://github.com/LTH14/mar.

## 1 Introduction

Autoregressive models are currently the *de facto* solution to generative models in natural language processing [38, 39, 3]. These models predict the next word or token in a sequence based on the previous words as input. Given the discrete nature of languages, the inputs and outputs of these models are in a *categorical*, *discrete-valued* space. This prevailing approach has led to a widespread belief that autoregressive models are inherently linked to discrete representations.

As a result, research on generalizing autoregressive models to *continuous-valued* domains—most notably, image generation—has intensely focused on discretizing the data [6, 13, 40]. A commonly adopted strategy is to train a discrete-valued tokenizer on images, which involves a finite vocabulary obtained by vector quantization (VQ) [51, 41]. Autoregressive models are then operated on the discrete-valued token space, analogous to their language counterparts.

In this work, we aim to address the following question: *Is it necessary for autoregressive models to be coupled with vector-quantized representations?* We note that the autoregressive nature, *i.e.*, "predicting next tokens based on previous ones", is independent of whether the values are discrete or continuous. What is needed is to model the per-token *probability distribution*, which can be measured by a loss function and used to draw samples from. Discrete-valued representations can be conveniently modeled by a categorical distribution, but it is not conceptually necessary. If alternative models for per-token probability distributions are presented, autoregressive models can be approached without vector quantization.

With this observation, we propose to model the *per-token* probability distribution by a diffusion procedure operating on continuous-valued domains. Our methodology leverages the principles of diffusion models [45, 24, 33, 10] for representing arbitrary probability distributions. Specifically, our method autoregressively predicts a vector $z$ for each token, which serves as a conditioning for a denoising network (*e.g.*, a small MLP). The denoising diffusion procedure enables us to represent an underlying distribution $p(x|z)$ for the output $x$ (Figure 1). This small denoising network is

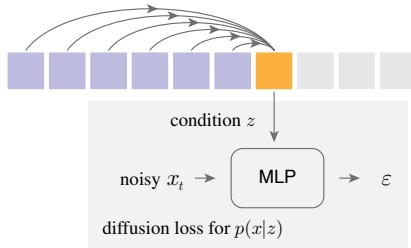

Figure 1: **Diffusion Loss**. Given a continuous-valued token $x$ to be predicted, the autoregressive model produces a vector $z$, which serves as the condition of a denoising diffusion network (a small MLP). This offers a way to model the probability distribution $p(x|z)$ of *this token*. This network is trained jointly with the autoregressive model by backpropagation. At inference time, with a predicted $z$, running the reverse diffusion procedure can sample a token following the distribution: $x \sim p(x|z)$. This method eliminates the need for discrete-valued tokenizers.

trained jointly with the autoregressive model, with continuous-valued tokens as the input and target. Conceptually, this small prediction head, applied to each token, behaves like a loss function for measuring the quality of $z$. We refer to this loss function as *Diffusion Loss*.

Our approach eliminates the need for discrete-valued tokenizers. Vector-quantized tokenizers are difficult to train and are sensitive to gradient approximation strategies [51, 41, 40, 27]. Their reconstruction quality often falls short compared to continuous-valued counterparts [42]. Our approach allows autoregressive models to enjoy the benefits of higher-quality, non-quantized tokenizers.

To broaden the scope, we further unify standard autoregressive (AR) models [13] and masked generative models [4, 29] into a generalized autoregressive framework (Figure 3). Conceptually, masked generative models predict *multiple* output tokens simultaneously in a *randomized* order, while still maintaining the autoregressive nature of "predicting next tokens based on known ones". This leads to a *masked autoregressive* (MAR) model that can be seamlessly used with Diffusion Loss.

We demonstrate by experiments the effectiveness of Diffusion Loss across a wide variety of cases, including AR and MAR models. It eliminates the need for vector-quantized tokenizers and consistently improves generation quality. Our loss function can be flexibly applied with different types of tokenizers. Further, our method enjoys the advantage of the fast speed of sequence models. Our MAR model with Diffusion Loss can generate at a rate of $< 0.3$ second per image while achieving a strong FID of $< 2.0$ on ImageNet $256{\times}256$. Our best model can approach 1.55 FID.

The effectiveness of our method reveals a largely uncharted realm of image generation: modeling the *interdependence* of tokens by autoregression, jointly with the *per-token* distribution by diffusion. This is in contrast with typical latent diffusion models [42, 37] in which the diffusion process models the joint distribution of all tokens. Given the effectiveness, speed, and flexibility of our method, we hope that the Diffusion Loss will advance autoregressive image generation and be generalized to other domains in future research.

## 2   Related Work

**Sequence Models for Image Generation**. Pioneering efforts on autoregressive image models [17, 50, 49, 36, 7, 6] operate on sequences of pixels. Autoregression can be performed by RNNs [50], CNNs [49, 7], and, most lately and popularly, Transformers [36, 6]. Motivated by language models, another series of works [51, 41, 13, 40] model images as discrete-valued tokens. Autoregressive [13, 40] and masked generative models [4, 29] can operate on the discrete-valued token space. But discrete tokenizers are difficult to train, which has recently drawn special focus [27, 54, 32].

Related to our work, the recent work on GIVT [48] also focuses on continuous-valued tokens in sequence models. GIVT and our work both reveal the significance and potential of this direction. In GIVT, the token distribution is represented by Gaussian mixture models. It uses a pre-defined number of mixtures, which can limit the types of distributions it can represent. In contrast, our method leverages the effectiveness of the diffusion process for modeling arbitrary distributions.

**Diffusion for Representation Learning**. The denoising diffusion process has been explored as a criterion for visual self-supervised learning. For example, DiffMAE [53] replaces the L2 loss in the original MAE [21] with a denoising diffusion decoder; DARL [30] trains autoregressive models with a denoising diffusion patch decoder. These efforts have been focused on representation learning, rather than image generation. In their scenarios, generating *diverse* images is not a goal; these methods have not presented the capability of generating new images from scratch.

**Diffusion for Policy Learning**. Our work is conceptually related to Diffusion Policy [8] in robotics. In those scenarios, the distribution of *taking an action* is formulated as a denoising process on the robot observations, which can be pixels or latents [8, 34]. In image generation, we can think of generating a token as an "action" to take. Despite this conceptual connection, the diversity of the generated samples in robotics is less of a core consideration than it is for image generation.

## 3 Method

In a nutshell, our image generation approach is a sequence model operated on a tokenized latent space [6, 13, 40]. But unlike previous methods that are based on vector-quantized tokenizers (*e.g.*, variants of VQ-VAE [51, 13]), we aim to use continuous-valued tokenizers (*e.g.*, [42]). We propose Diffusion Loss that makes sequence models compatible with continuous-valued tokens.

### 3.1 Rethinking Discrete-Valued Tokens

To begin with, we revisit the roles of discrete-valued tokens in autoregressive generation models. Denote as $x$ the ground-truth token to be predicted at the next position. With a discrete tokenizer, $x$ can be represented as an integer: $0 \leq x < K$, with a vocabulary size $K$. The autoregressive model produces a continuous-valued $D$-dim vector $z \in \mathbb{R}^D$, which is then projected by a $K$-way classifier matrix $W \in \mathbb{R}^{K \times D}$. Conceptually, this formulation models a *categorical probability distribution* in the form of $p(x|z) = \text{softmax}(Wz)$.

In the context of generative modeling, this probability distribution must exhibit two essential properties. (i) A **loss function** that can measure the difference between the estimated and true distributions. In the case of categorical distribution, this can be simply done by the cross-entropy loss. (ii) A **sampler** that can draw samples from the distribution $x \sim p(x|z)$ at inference time. In the case of categorical distribution, this is often implemented as drawing a sample from $p(x|z) = \text{softmax}(Wz/\tau)$, in which $\tau$ is a temperature that controls the diversity of the samples. Sampling from a categorical distribution can be approached by the Gumbel-max method [18] or inverse transform sampling.

This analysis suggests that discrete-valued tokens are *not* necessary for autoregressive models. Instead, it is the requirement of modeling a distribution that is essential. A discrete-valued token space implies a categorical distribution, whose loss function and sampler are simple to define. What we actually need are a loss function and its corresponding sampler for distribution modeling.

### 3.2 Diffusion Loss

Denoising diffusion models [24] offer an effective framework to model arbitrary distributions. But unlike common usages of diffusion models for representing the joint distribution of all pixels or all tokens, in our case, the diffusion model is for representing the distribution *for each token*.

Consider a continuous-valued vector $x \in \mathbb{R}^d$, which denotes the ground-truth token to be predicted at the next position. The autoregressive model produces a vector $z \in \mathbb{R}^D$ at this position. Our goal is to model a probability distribution of $x$ conditioned on $z$, that is, $p(x|z)$. The loss function and sampler can be defined following the diffusion models [24, 33, 10], described next.

**Loss function**. Following [24, 33, 10], the loss function of an underlying probability distribution $p(x|z)$ can be formulated as a denoising criterion:

$$\mathcal{L}(z, x) = \mathbb{E}_{\varepsilon, t} \left[ \|\varepsilon - \varepsilon_\theta(x_t|t, z)\|^2 \right]. \tag{1}$$

Here, $\varepsilon \in \mathbb{R}^d$ is a noise vector sampled from $\mathcal{N}(\mathbf{0}, \mathbf{I})$. The noise-corrupted vector $x_t$ is $x_t = \sqrt{\bar{\alpha}_t}x + \sqrt{1 - \bar{\alpha}_t}\varepsilon$, where $\bar{\alpha}_t$ defines a noise schedule [24, 33]. $t$ is a time step of the noise schedule. The noise estimator $\varepsilon_\theta$, parameterized by $\theta$, is a small MLP network (see Sec. 4). The notation $\varepsilon_\theta(x_t|t, z)$ means that this network takes $x_t$ as the input, and is *conditional* on both $t$ and $z$. As per [46, 47], Eqn. (1) conceptually behaves like a form of score matching: it is related to a loss function concerning the score function of $p(x|z)$, that is, $\nabla \log_x p(x|z)$. Diffusion Loss is a *parameterized* loss function, in the same vein as the adversarial loss [15] or perceptual loss [56].

It is worth noticing that the conditioning vector $z$ is produced by the autoregressive network: $z = f(\cdot)$, as we will discuss later. The gradient of $z = f(\cdot)$ is propagated from the loss function in Eqn. (1). Conceptually, Eqn. (1) defines a loss function for training the network $f(\cdot)$.

We note that the expectation $\mathbb{E}_{\varepsilon,t}[\cdot]$ in Eqn. (1) is over $t$, for any given $z$. As our denoising network is small, we can sample $t$ multiple times for any given $z$. This helps improve the utilization of the loss function, without recomputing $z$. We sample $t$ by 4 times during training for each image.

**Sampler**. At inference time, it is required to draw samples from the distribution $p(x|z)$. Sampling is done via a reverse diffusion procedure [24]: $x_{t-1} = \frac{1}{\sqrt{\alpha_t}}\left(x_t - \frac{1-\alpha_t}{\sqrt{1-\bar{\alpha}_t}}\varepsilon_\theta(x_t|t,z)\right) + \sigma_t\delta$. Here $\delta$ is sampled from the Gaussian distribution $\mathcal{N}(\mathbf{0},\mathbf{I})$ and $\sigma_t$ is the noise level at time step $t$. Starting with $x_T \sim \mathcal{N}(\mathbf{0},\mathbf{I})$, this procedure produces a sample $x_0$ such that $x_0 \sim p(x|z)$ [24].

When using categorical distributions (Sec. 3.1), autoregressive models can enjoy the benefit of having a **temperature** $\tau$ for controlling sample diversity. In fact, existing literature, in both languages and images, has shown that temperature plays a critical role in autoregressive generation. It is desired for the diffusion sampler to offer a temperature counterpart. We adopt the temperature sampling presented in [10]. Conceptually, with temperature $\tau$, one may want to sample from the (renormalized) probability of $p(x|z)^{\frac{1}{\tau}}$, whose score function is $\frac{1}{\tau}\nabla \log_x p(x|z)$. In practice, [10] suggests to either divide $\varepsilon_\theta$ by $\tau$, or scale the noise by $\tau$. We adopt the latter option: we scale $\sigma_t\delta$ in the sampler by $\tau$. Intuitively, $\tau$ controls the sample diversity by adjusting the noise variance.

### 3.3 Diffusion Loss for Autoregressive Models

Next, we describe the autoregressive model with Diffusion Loss for image generation. Given a sequence of tokens $\{x^1, x^2, ..., x^n\}$ where the superscript $1 \le i \le n$ specifies an order, autoregressive models [17, 50, 49, 36, 7, 6] formulate the generation problem as "next token prediction":

$$p(x^1, ..., x^n) = \prod_{i=1}^{n} p(x^i \mid x^1, ..., x^{i-1}). \tag{2}$$

A network is used to represent the conditional probability $p(x^i \mid x^1, ..., x^{i-1})$. In our case, $x^i$ can be continuous-valued. We can rewrite this formulation in two parts. We first produce a conditioning vector $z^i$ by a network (*e.g.*, Transformer [52]) operating on previous tokens: $z^i = f(x^1, ..., x^{i-1})$. Then, we model the probability of the next token by $p(x^i|z^i)$. Diffusion Loss in Eqn. (1) can be applied on $p(x^i|z^i)$. The gradient is backpropagated to $z^i$ for updating the parameters of $f(\cdot)$.

### 3.4 Unifying Autoregressive and Masked Generative Models

We show that *masked generative models*, *e.g.*, MaskGIT [4] and MAGE [29], can be generalized under the broad concept of autoregression, *i.e.*, next token prediction.

**Bidirectional attention can perform autoregression**. The concept of autoregression is orthogonal to network architectures: autoregression can be done by RNNs [50], CNNs [49, 7], and Transformers [38, 36, 6]. When using Transformers, although autoregressive models are popularly implemented by *causal* attention, we show that they can also be done by *bidirectional* attention. See Figure 2. Note that the goal of autoregression is to *predict the next token given the previous tokens*; it does *not* constrain how the previous tokens communicate with the next token.

We can adopt the bidirectional attention implementation as done in Masked Autoencoder (MAE) [21]. See Figure 2(b). Specifically, we first apply an MAE-style encoder[1] on the known tokens (with positional embedding [52]). Then we concatenate the encoded sequence with mask tokens (with positional embedding added again), and map this sequence with an MAE-style decoder. The positional embedding on the mask tokens can let the decoder know at which positions are to be predicted. Unlike causal attention, here the loss is computed only on the unknown tokens [21].

With the MAE-style trick, we allow *all* known tokens to see each other, and also allow all unknown tokens to see all known tokens. This *full attention* introduces better communication across tokens than causal attention. At inference time, we can generate tokens (one or more per step) using this bidirectional formulation, which is a form of autoregression. As a compromise, we cannot use the key-value (kv) cache [44] of causal attention to speed up inference. But as we can generate multiple tokens together, we can reduce generation steps to speed up inference. Full attention across tokens can significantly improve the quality and offer a better speed/accuracy trade-off.

---

[1]Here the terminology of encoder/decoder is in the sense of a general Autoencoder, following MAE [21]. It is not related to whether the computation is casual/bidirectional in Transformers [52].

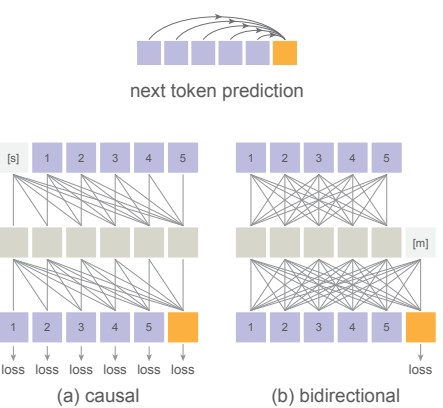

Figure 2: **Bidirectional attention can do autoregression**. In contrast to conventional wisdom, the broad concept of "autoregression" (next token prediction) can be done by either causal or bidirectional attention. (a) **Causal** attention restricts each token to attend only to current/previous tokens. With input shifted by one start token [s], it is valid to compute loss on *all* tokens at training time. (b) **Bidirectional** attention allows each token to see *all* tokens in the sequence. Following MAE [21], mask tokens [m] are applied in a middle layer, with positional embedding added. This setup only computes loss on unknown tokens, but it allows for full attention capabilities across the sequence, enabling *better* communication across tokens. This setup can generate tokens one by one at inference time, which is a form of autoregression. It also allows us to predict multiple tokens simultaneously.

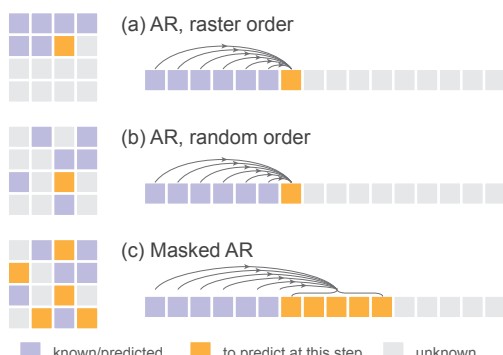

Figure 3: **Generalized Autoregressive Models.** (a) A standard, raster-order autoregressive model predicts one next token based on the previous tokens. (b) A random-order autoregressive model predicts the next token given a random order. It behaves like randomly masking out tokens and then predicting one. (c) A Masked Autoregressive (MAR) model predicts multiple tokens simultaneously given a random order, which is conceptually analogous to masked generative models [4, 29]. In all cases, the prediction of one step can be done by causal or bidirectional attention (Figure 2).

**Autoregressive models in random orders**. To connect to masked generative models [4, 29], we consider an autoregressive variant in random orders. The model is given a randomly permuted sequence. This random permutation is different for each sample. See Figure 3(b). In this case, the position of the next token to be predicted needs to be accessible to the model. We adopt a strategy similar to MAE [21]: we add positional embedding (that corresponds to the unshuffled positions) to the decoder layers, which can tell what positions to predict. This strategy is applicable for both causal and bidirectional versions.

As shown in Figure 3 (b)(c), random-order autoregression behaves like a special form of masked generation, in which one token is generated at a time. We elaborate on this as follows.

**Masked autoregressive models**. In masked generative modeling [4, 29], the models predict a random subset of tokens based on known/predicted tokens. This can be formulated as permuting the token sequence by a random order, and then predicting *multiple* tokens based on previous tokens. See Figure 3(c). Conceptually, this is an autoregressive procedure, which can be written as estimating the conditional distribution: $p(\{x^i, x^{i+1}..., x^j\} \mid x^1, ..., x^{i-1})$, where multiple tokens $\{x^i, x^{i+1}..., x^j\}$ are to be predicted ($i \leq j$). We can write this autoregressive model as:

$$p(x^1, ..., x^n) = p(X^1, ..., X^K) = \prod_k^K p(X^k \mid X^1, ..., X^{k-1}). \tag{3}$$

Here, $X^k = \{x^i, x^{i+1}..., x^j\}$ is *a set of tokens* to be predicted at the $k$-th step, with $\cup_k X^k = \{x^1, ..., x^n\}$. In this sense, this is essentially "*next set-of-tokens prediction*", and thus is also a general form of autoregression. We refer to this variant as Masked Autoregressive (MAR) models. MAR is a random-order autoregressive model that can predict multiple tokens simultaneously.

MAR is conceptually related to MAGE [29]. However, MAR samples tokens by a temperature $\tau$ applied on the probability distribution of *each* token (which is the standard practice in generative language models like GPT). In contrast, MAGE (following MaskGIT [4]) applies a temperature for sampling the *locations* of the tokens to be predicted: this is not a fully randomized order, which creates a gap between training-time and inference-time behavior.

# 4 Implementation

This section describes our implementation. We note that the concepts introduced in this paper are general and not limited to specific implementations. More detailed specifics are in Appendix B.

## 4.1 Diffusion Loss

**Diffusion Process**. Our diffusion process follows [33]. Our noise schedule has a cosine shape, with 1000 steps at training time; at inference time, it is resampled with fewer steps (by default, 100) [33]. Our denoising network predicts the noise vector $\varepsilon$ [24]. The loss can optionally include the variational lower bound term $\mathcal{L}_{\text{vlb}}$ [33]. Diffusion Loss naturally supports classifier-free guidance (CFG) [23] (detailed in Appendix B).

**Denoising MLP**. We use a small MLP consisting of a few residual blocks [20] for denoising. Each block sequentially applies a LayerNorm (LN) [1], a linear layer, SiLU [12], and another linear layer, merging with a residual connection. By default, we use 3 blocks and a width of 1024 channels. The denoising MLP is conditioned on a vector $z$ produced by the AR/MAR model (see Figure 1). The vector $z$ is added to the time embedding of the noise schedule time-step $t$, which serves as the condition of the MLP in the LN layers via AdaLN [37].

## 4.2 Autoregressive and Masked Autoregressive Image Generation

**Tokenizer**. We use the publicly available tokenizers provided by LDM [42]. Our experiments will involve their VQ-16 and KL-16 versions [42]. VQ-16 is a VQ-GAN [13], *i.e.*, VQ-VAE [51] with GAN loss [15] and perceptual loss [56]; KL-16 is its counterpart regularized by Kullback–Leibler (KL) divergence, *without* vector quantization. 16 denotes the tokenizer strides.

**Transformer**. Our architecture follows the Transformer [52] implementation in ViT [11]. Given a sequence of tokens from a tokenizer, we add positional embedding [52] and append the class tokens [cls]; then we process the sequence by a Transformer. By default, our Transformer has 32 blocks and a width of 1024, which we refer to as the Large size or -L ($\sim$400M parameters).

**Autoregressive baseline**. Causal attention is implemented following the common practice of GPT [38] (Figure 2(a)). The input sequence is shifted by one token (here, [cls]). Triangular masking [52] is applied to the attention matrix. At inference time, temperature ($\tau$) sampling is applied. We use kv-cache [44] for efficient inference.

**Masked autoregressive models**. With bidirectional attention (Figure 2(b)), we can predict any number of unknown tokens given any number of known tokens. At training time, we randomly sample a masking ratio [21, 4, 29] in [0.7, 1.0]: *e.g.*, 0.7 means 70% tokens are unknown. Because the sampled sequence can be very short, we always pad 64 [cls] tokens at the start of the encoder sequence, which improves the stability and capacity of our encoding. As in Figure 2, mask tokens [m] are introduced in the decoder, with positional embedding added. For simplicity, unlike [21], we let the encoder and decoder have the same size: each has half of all blocks (*e.g.*, 16 in MAR-L).

At inference, MAR performs "next set-of-tokens prediction". It progressively reduces the masking ratio from 1.0 to 0 with a cosine schedule [4, 29]. By default, we use 64 steps in this schedule. Temperature ($\tau$) sampling is applied. Unlike [4, 29], MAR always uses fully randomized orders.

# 5 Experiments

We experiment on ImageNet [9] at a resolution of 256×256. We evaluate FID [22] and IS [43], and provide Precision and Recall as references following common practice [10]. We follow the evaluation suite provided by [10].

## 5.1 Properties of Diffusion Loss

**Diffusion Loss *vs*. Cross-entropy Loss**. We first compare continuous-valued tokens with Diffusion Loss and standard discrete-valued tokens with cross-entropy loss (Table 1). For fair comparisons, the tokenizers ("VQ-16" and "KL-16") are both downloaded from the LDM codebase [42]. These are popularly used tokenizers (*e.g.*, [13, 42, 37]).

Table 1: **Diffusion Loss vs. Cross-entropy Loss**. The tokenizers are VQ-16 (discrete) and KL-16 (continuous), both from the LDM codebase [42] for fair comparisons. Diffusion Loss, with *continuous*-valued tokens, is better than its cross-entropy counterpart with *discrete*-valued tokens, consistently observed across all variants of AR and MAR. All entries are implemented by us under the same setting: AR/MAR-L ($\sim$400M parameters), 400 epochs, ImageNet 256×256.

| variant | order | direction | # preds | loss | w/o CFG FID↓ | w/o CFG IS↑ | w/ CFG FID↓ | w/ CFG IS↑ |
|---|---|---|---|---|---|---|---|---|
| AR | raster | causal | 1 | CrossEnt | 19.58 | 60.8 | 4.92 | 227.3 |
| | | | | **Diff Loss** | **19.23** | **62.3** | **4.69** | **244.6** |
| MAR | rand | causal | 1 | CrossEnt | 16.22 | 81.3 | 4.36 | 222.7 |
| | | | | **Diff Loss** | **13.07** | **91.4** | **4.07** | **232.4** |
| MAR | rand | bidirect | 1 | CrossEnt | 8.75 | 149.6 | 3.50 | 280.9 |
| | | | | **Diff Loss** | **3.43** | **203.1** | **1.84** | **292.7** |
| MAR (default) | rand | bidirect | >1 | CrossEnt | 8.79 | 146.1 | 3.69 | 278.4 |
| | | | | **Diff Loss** | **3.50** | **201.4** | **1.98** | **290.3** |

Table 2: **Flexibility of Diffusion Loss**. Diffusion Loss can support different types of tokenizers. **(i)** VQ tokenizers: we treat the continuous-valued latent before VQ as the tokens. **(ii)** Tokenizers with a *mismatched* stride (here, 8): we group 2×2 tokens into a new token for sequence modeling. **(iii)** Consistency Decoder [35], a non-VQ tokenizer of a different decoder architecture. Here, rFID denotes the reconstruction FID of the tokenizer on the ImageNet training set. Settings in this table for all entries: MAR-L, 400 epochs, ImageNet 256×256. [†]: This tokenizer is trained by us on ImageNet using [42]'s code; the original ones from [42] were trained on OpenImages.

| loss | tokenizer src | tokenizer arch | # tokens raw | # tokens seq | rFID↓ | w/o CFG FID↓ | w/o CFG IS↑ | w/ CFG FID↓ | w/ CFG IS↑ |
|---|---|---|---|---|---|---|---|---|---|
| | [42] | VQ-16 | $16^2$ | $16^2$ | 5.87 | 7.82 | 151.7 | 3.64 | 258.5 |
| | [42] | KL-16 | $16^2$ | $16^2$ | 1.43 | 3.50 | 201.4 | 1.98 | 290.3 |
| Diff Loss | [42] | KL-8 | $32^2$ | $16^2$ | 1.20 | 4.33 | 180.0 | 2.05 | 283.9 |
| | [35] | Consistency | $32^2$ | $16^2$ | 1.30 | 5.76 | 170.6 | 3.23 | 271.0 |
| | [42][†] | KL-16 | $16^2$ | $16^2$ | 1.22 | 2.85 | 214.0 | 1.97 | 291.2 |

The comparisons are in four variants of AR/MAR. As shown in Table 1, Diffusion Loss consistently outperforms the cross-entropy counterpart in all cases. Specifically, in MAR (*e.g.*, the default), using Diffusion Loss can reduce FID by relatively ~50%-60%. This is because the continuous-valued KL-16 has smaller compression loss than VQ-16 (discussed next in Table 2), and also because a diffusion process models distributions more effectively than categorical ones.

In the following ablations, unless specified, we follow the "default" MAR setting in Table 1.

**Flexibility of Diffusion Loss**. One significant advantage of Diffusion Loss is its flexibility with various tokenizers. We compare several publicly available tokenizers in Table 2.

Diffusion Loss can be easily used even given a VQ tokenizer. We simply treat the continuous-valued latent before the VQ layer as the tokens. This variant gives us 7.82 FID (w/o CFG), compared favorably with 8.79 FID (Table 1) of cross-entropy loss using the *same* VQ tokenizer. This suggests the better capability of diffusion for modeling distributions.

This variant also enables us to compare the VQ-16 and KL-16 tokenizers using *the same loss*. As shown in Table 2, VQ-16 has a much *worse* reconstruction FID (rFID) than KL-16, which consequently leads to a much worse generation FID (*e.g.*, 7.82 *vs.* 3.50 in Table 2).

Interestingly, Diffusion Loss also enables us to use tokenizers with *mismatched* strides. In Table 2, we study a KL-8 tokenizer whose stride is 8 and output sequence length is 32×32. Without increasing the sequence length of the generator, we group 2×2 tokens into a new token. Despite the mismatch, we are able to obtain decent results, *e.g.*, KL-8 gives us 2.05 FID, *vs.* KL-16's 1.98 FID. Further, this property allows us to investigate other tokenizers, *e.g.*, Consistency Decoder [35], a non-VQ tokenizer of a different architecture/stride designed for different goals.

For comprehensiveness, we also train a KL-16 tokenizer on ImageNet using the code of [42], noting that the original KL-16 in [42] was trained on OpenImages [28]. The comparison is in the last row of Table 2. We use this tokenizer in the following explorations.

| MLP | | w/o CFG | | w/ CFG | | |
| width | params | FID↓ | IS↑ | FID↓ | IS↑ | inference time |
|---|---|---|---|---|---|---|
| 256 | 2M | 3.47 | 195.3 | 2.45 | 274.0 | 0.286 s / im. |
| 512 | 6M | 3.24 | 199.1 | 2.11 | 281.0 | 0.288 s / im. |
| 1024 | 21M | 2.85 | 214.0 | 1.97 | 291.2 | 0.288 s / im. |
| 1536 | 45M | 2.93 | 207.6 | 1.91 | 289.3 | 0.291 s / im. |

Table 3: **Denoising MLP in Diffusion Loss**. The denoising MLP is small and efficient. Here, the inference time involves the entire generation model, and the Transformer's size is 407M. Settings: MAR-L, 400 epochs, ImageNet 256×256, 3 MLP blocks.

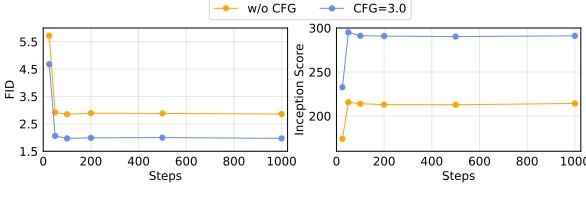

Figure 4: **Sampling steps of Diffusion Loss**. We show the FID (left) and IS (right) w.r.t. the number of diffusive sampling steps. Using 100 steps is sufficient to achieve a strong generation quality.

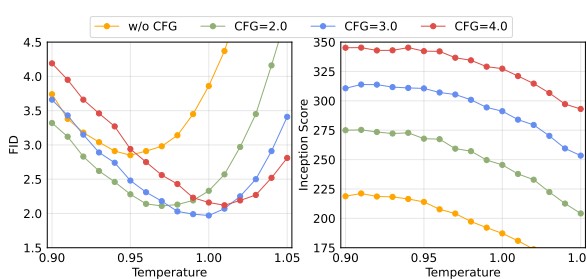

Figure 5: **Temperature of Diffusion Loss**. Temperature $\tau$ has clear influence on both FID (left) and IS (right). Just like the temperature in *discrete-valued* autoregression, the temperature here also plays a critical role in *continuous-valued* autoregression.

**Denoising MLP in Diffusion Loss**. We investigate the denoising MLP in Table 3. Even a very small MLP (*e.g.*, 2M) can lead to competitive results. As expected, increasing the MLP width helps improve the generation quality; we have explored increasing the depth and had similar observations. Note that our default MLP size (1024 width, 21M) adds only ~5% extra parameters to the MAR-L model. During inference, the diffusion sampler has a decent cost of ~10% overall running time. Increasing the MLP width has negligible extra cost in our implementation (Table 3), partially because the main overhead is not about computation but memory communication.

**Sampling Steps of Diffusion Loss**. Our diffusion process follows the common practice of DDPM [24, 10]: we train with a 1000-step noise schedule but inference with fewer steps. Figure 4 shows that using 100 diffusion steps at inference is sufficient to achieve a strong generation quality.

**Temperature of Diffusion Loss**. In the case of cross-entropy loss, the temperature is of central importance. Diffusion Loss also offers a temperature counterpart for controlling the diversity and fidelity. Figure 5 shows the influence of the temperature $\tau$ in the diffusion sampler (see Sec. 3.2) at inference time. The temperature $\tau$ plays an important role in our models, similar to the observations on cross-entropy-based counterparts (note that the cross-entropy results in Table 1 are with their optimal temperatures).

## 5.2 Properties of Generalized Autoregressive Models

**From AR to MAR**. Table 1 is also a comparison on the AR/MAR variants, which we discuss next. First, replacing the raster order in AR with *random* order has a significant gain, *e.g.*, reducing FID from 19.23 to 13.07 (w/o CFG). Next, replacing the causal attention with the *bidirectional* counterpart leads to another massive gain, *e.g.*, reducing FID from 13.07 to 3.43 (w/o CFG).

The random-order, bidirectional AR is essentially a form of MAR that predicts one token at a time. Predicting *multiple* tokens ('>1') at each step can effectively reduce the number of autoregressive steps. In Table 1, we show that the MAR variant with 64 steps slightly trades off generation quality. A more comprehensive trade-off comparison is discussed next.

**Speed/accuracy Trade-off**. Following MaskGIT [4], our MAR enjoys the flexibility of predicting multiple tokens at a time. This is controlled by the number of autoregressive steps at inference time. Figure 6 plots the speed/accuracy trade-off. MAR has a better trade-off than its AR counterpart, noting that AR is with the efficient kv-cache.

With Diffusion Loss, MAR also shows a favorable trade-off in comparison with the recently popular Diffusion Transformer (DiT) [37]. As a latent diffusion model, DiT models the interdependence

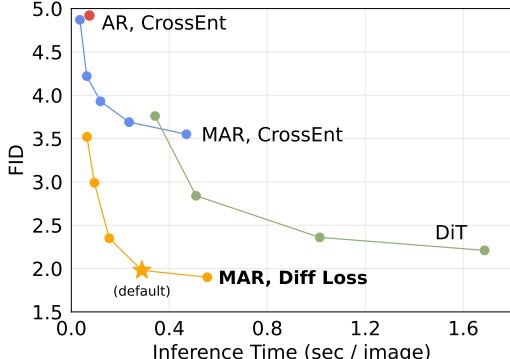

Figure 6: **Speed/accuracy trade-off** of the generation process. For MAR, a curve is obtained by different autoregressive steps (8 to 128). For DiT, a curve is obtained by different diffusion steps (50, 75, 150, 250) using its official code. We compare our implementation of AR and MAR. AR is with kv-cache for fast inference. AR/MAR model size is L and DiT model size is DiT-XL. The star marker denotes our default MAR setting used in other ablations. We benchmark FID and speed on ImageNet 256×256 using one A100 GPU with a batch size of 256.

Table 4: **System-level comparison** on ImageNet 256×256 conditional generation. Diffusion Loss enables Masked Autoregression to achieve leading results in comparison with previous systems.
[†]: LDM operates on continuous-valued tokens, though this result uses a quantized tokenizer.

| | #params | w/o CFG | | | | w/ CFG | | | |
|---|---|---|---|---|---|---|---|---|---|
| | | FID↓ | IS↑ | Pre.↑ | Rec.↑ | FID↓ | IS↑ | Pre.↑ | Rec.↑ |
| *pixel-based* | | | | | | | | | |
| ADM [10] | 554M | 10.94 | 101.0 | 0.69 | 0.63 | 4.59 | 186.7 | 0.82 | 0.52 |
| VDM++ [26] | 2B | 2.40 | 225.3 | - | - | 2.12 | 267.7 | - | - |
| *vector-quantized tokens* | | | | | | | | | |
| Autoreg. w/ VQGAN [13] | 1.4B | 15.78 | 78.3 | - | - | - | - | - | - |
| MaskGIT [4] | 227M | 6.18 | 182.1 | 0.80 | 0.51 | - | - | - | - |
| MAGE [29] | 230M | 6.93 | 195.8 | - | - | - | - | - | - |
| MAGVIT-v2 [55] | 307M | 3.65 | 200.5 | - | - | 1.78 | **319.4** | - | - |
| *continuous-valued tokens* | | | | | | | | | |
| LDM-4[†] [42] | 400M | 10.56 | 103.5 | 0.71 | 0.62 | 3.60 | 247.7 | 0.87 | 0.48 |
| U-ViT-H/2-G [2] | 501M | - | - | - | - | 2.29 | 263.9 | 0.82 | 0.57 |
| DiT-XL/2 [37] | 675M | 9.62 | 121.5 | 0.67 | 0.67 | 2.27 | 278.2 | 0.83 | 0.57 |
| DiffiT [19] | - | - | - | - | - | 1.73 | 276.5 | 0.80 | 0.62 |
| MDTv2-XL/2 [14] | 676M | 5.06 | 155.6 | 0.72 | 0.66 | 1.58 | 314.7 | 0.79 | 0.65 |
| GIVT [48] | 304M | 5.67 | - | 0.75 | 0.59 | 3.35 | - | 0.84 | 0.53 |
| MAR-B, Diff Loss | 208M | 3.48 | 192.4 | 0.78 | 0.58 | 2.31 | 281.7 | 0.82 | 0.57 |
| MAR-L, Diff Loss | 479M | 2.60 | 221.4 | 0.79 | 0.60 | 1.78 | 296.0 | 0.81 | 0.60 |
| MAR-H, Diff Loss | 943M | **2.35** | **227.8** | 0.79 | 0.62 | **1.55** | 303.7 | 0.81 | 0.62 |

across *all* tokens by the diffusion process. The speed/accuracy trade-off of DiT is mainly controlled by its diffusion steps. Unlike our diffusion process on a small MLP, the diffusion process of DiT involves the *entire* Transformer architecture. Our method is more accurate and faster. Notably, our method can generate at a rate of $< 0.3$ second per image with a strong FID of $< 2.0$.

### 5.3 Benchmarking with Previous Systems

We compare with the leading systems in Table 4. We explore various model sizes (see Appendix B) and train for 800 epochs. Similar to autoregressive language models [3], we observe encouraging scaling behavior. Further investigation into scaling could be promising. Regarding metrics, we report 2.35 FID *without* CFG, largely outperforming other token-based methods. Our best entry has 1.55 FID and compares favorably with leading systems. Figure 7 shows qualitative results.

## 6 Discussion and Conclusion

The effectiveness of Diffusion Loss on various autoregressive models suggests new opportunities: modeling the *interdependence* of tokens by autoregression, jointly with the *per-token* distribution by diffusion. This is unlike the common usage of diffusion that models the joint distribution of all tokens. Our strong results on image generation suggest that autoregressive models or their extensions are powerful tools beyond language modeling. These models do not need to be constrained by vector-quantized representations. We hope our work will motivate the research community to explore sequence models with continuous-valued representations in other domains.

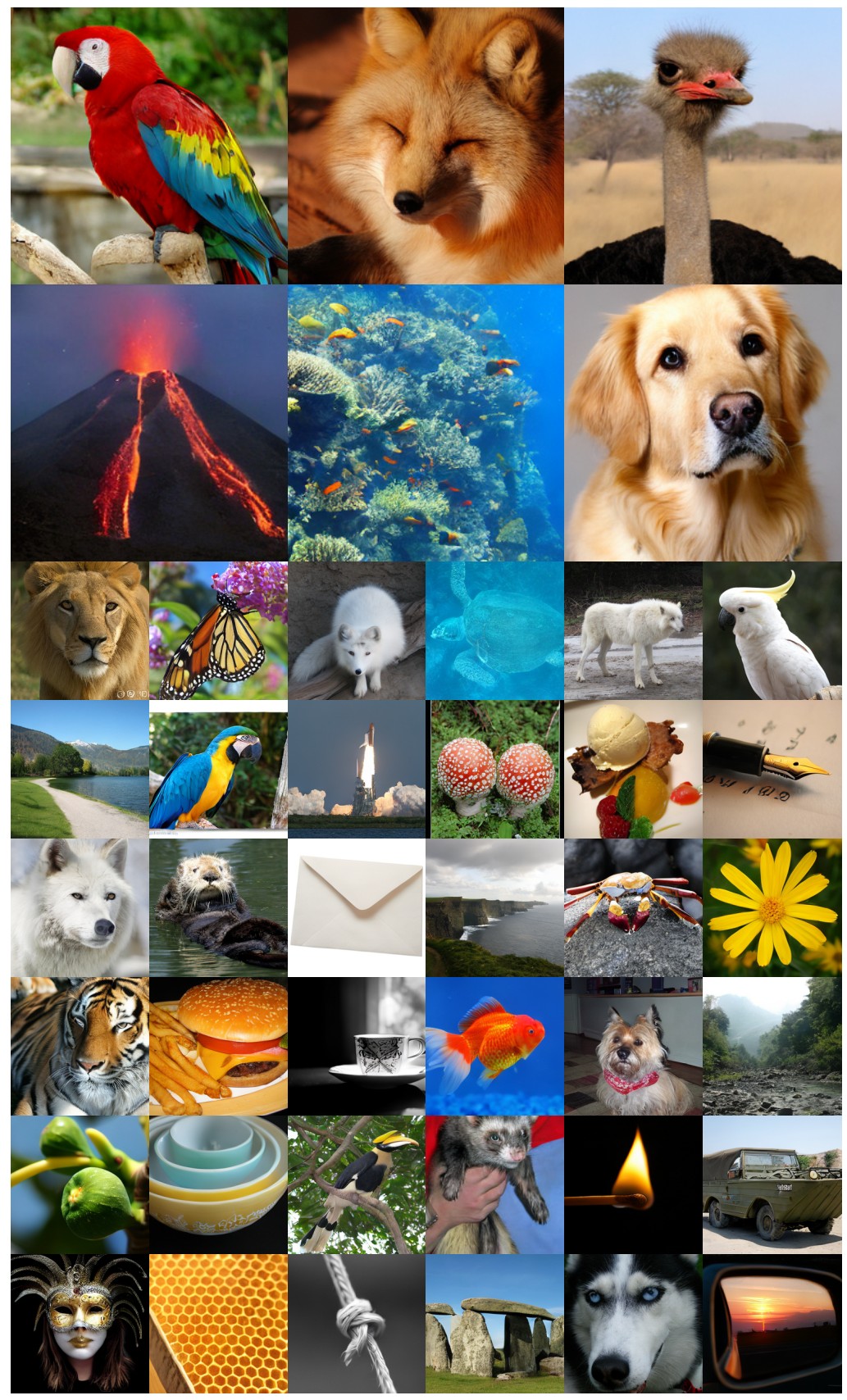

Figure 7: **Qualitative Results.** We show selected examples of class-conditional generation on ImageNet 256×256 using MAR-H with Diffusion Loss.

**Acknowledgements.** Tianhong Li was supported by the Mathworks Fellowship during this project. We thank Congyue Deng and Xinlei Chen for helpful discussion. We thank Google TPU Research Cloud (TRC) for granting us access to TPUs, and Google Cloud Platform for supporting GPU resources.

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

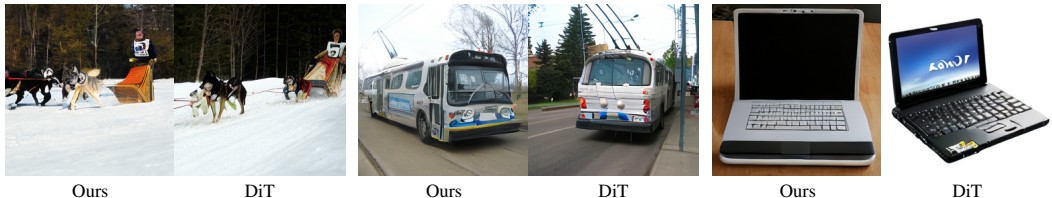

| Ours | DiT | Ours | DiT | Ours | DiT |

Figure 8: **Failure cases**. Similar to existing methods, our system can produce results with noticeable artifacts. For each pair, we show MAR-H and DiT-XL's results of the same class. The leftmost example of DiT is taken from their paper [37]; the others are obtained from their official code.

## A    Limitations and Broader Impacts

**Limitations.** Beyond demonstrating the potential of our method for image generation, this paper acknowledges its limitations.

First of all, our image generation system can produce images with noticeable artifacts (Figure 8). This limitation is commonly observed in existing methods, especially when trained on controlled, academic data (*e.g.*, ImageNet). Research-driven models trained on ImageNet still have a noticeable gap in visual quality in comparison with commercial models trained on massive data.

Second, our image generation system relies on existing pre-trained tokenizers. The quality of our system can be limited by the quality of these tokenizers. Pre-training better tokenizers is beyond the scope of this paper. Nevertheless, we hope our work will make it easier to use continuous-valued tokenizers to be developed in the future.

Last, we note that given the limited computational resources, we have primarily tested our method on the ImageNet benchmark. Further validation is needed to assess the scalability and robustness of our approach in more diverse and real-world scenarios.

**Broader Impacts.** Our primary aim is to advance the fundamental research on generative models, and we believe it will be beneficial to this field. An immediate application of our method is to extend it to large visual generation models, *e.g.*, text-to-image or text-to-video generation. Our approach has the potential to significantly reduce the training and inference cost of these large models. At the same time, our method may suggest the opportunity to replace traditional loss functions with Diffusion Loss in many applications. On the negative side, our method learns statistics from the training dataset, and as such may reflect the bias in the data; the image generation system may be misused to generate disinformation, which warrants further consideration.

## B    Additional Implementation Details

**Classifier-free guidance (CFG)**. To support CFG [23], at training time, the class condition is replaced with a dummy class token for 10% of the samples [23]. At inference time, the model is run with the given class token and the dummy token, providing two outputs $z_c$ and $z_u$. The predicted noise $\varepsilon$ is then modified [23] as: $\varepsilon = \varepsilon_\theta(x_t|t, z_u) + \omega \cdot (\varepsilon_\theta(x_t|t, z_c) - \varepsilon_\theta(x_t|t, z_u))$, where $\omega$ is the guidance scale. At inference time, we use a CFG schedule following [5]. We sweep the optimal guidance scale and temperature combination for each model.

**Training**. By default, the models are trained using the AdamW optimizer [31] for 400 epochs. The weight decay and momenta for AdamW are 0.02 and (0.9, 0.95). We use a batch size of 2048 and a learning rate (lr) of 8e-4. Our models with Diffusion Loss are trained with a 100-epoch linear lr warmup [16], followed by a *constant* [37] lr schedule. The cross-entropy counterparts are trained with a cosine lr schedule, which works better for them. Following [37, 25], we maintain the exponential moving average (EMA) of the model parameters with a momentum of 0.9999.

**Implementation Details of Table** 4. To explore our method's scaling behavior, we study three model sizes described as follows. In addition to MAR-L, we explore a smaller model (MAR-B) and a larger model (MAR-H). MAR-B, -L, and -H respectively have 24, 32, 40 Transformer blocks and a width of 768, 1024, and 1280. In Table 4 specifically, the denoising MLP respectively has 6, 8, 12 blocks and a width of 1024, 1280, and 1536. The training length is increased to 800 epochs. At inference time, we run 256 autoregressive steps to achieve the best results.

**Pseudo-code of Diffusion Loss**. See Algorithm 1.

**Algorithm 1** Diffusion Loss: PyTorch-like Pseudo-code

```
class DiffusionLoss(nn.Module)
    def __init__(depth, width):
        # SimpleMLP takes in x_t, timestep, and condition, and outputs predicted noise.
        self.net = SimpleMLP(depth, width)

        # GaussianDiffusion offers forward and backward functions q_sample and p_sample.
        self.diffusion = GaussianDiffusion()

    # Given condition z and ground truth token x, compute loss
    def loss(self, z, x):
        # sample random noise and timestep
        noise = torch.randn(x.shape)
        timestep = torch.randint(0, self.diffusion.num_timesteps, x.size(0))

        # sample x_t from x
        x_t = self.diffusion.q_sample(x, timestep, noise)

        # predict noise from x_t
        noise_pred = self.net(x_t, timestep, z)

        # L2 loss
        loss = ((noise_pred - noise) ** 2).mean()

        # optional: loss += loss_vlb

        return loss

    # Given condition and noise, sample x using reverse diffusion process
    def sample(self, z, noise):
        x = noise
        for t in list(range(self.diffusion.num_timesteps))[::-1]:
            x = self.diffusion.p_sample(self.net, x, t, z)
        return x
```

**Pseudo-code illustrating the concept of Diffusion Loss**. Here the conditioning vector $z$ is the output from the AR/MAR model. The gradient is backpropagated to $z$. For simplicity, here we omit the code for inference rescheduling, temperature and the loss term for variational lower bound [10], which can be easily incorporated.

**Compute Resources**. Our training is mainly done on 16 servers with 8 V100 GPUs each. Training a 400 epochs MAR-L model takes ∼2.6 days on these GPUs. As a comparison, training a DiT-XL/2 and LDM-4 model for the same number of epochs on this cluster takes 4.6 and 9.5 days, respectively.

## C  Comparison between MAR and MAGE

MAR (regardless of the loss used) is conceptually related to MAGE [29]. Besides implementation differences (*e.g.*, architecture specifics, hyper-parameters), a major conceptual difference between MAR and MAGE is in the scanning order at inference time. In MAGE, following MaskGIT [4], the locations of the next tokens to be predicted are determined *on-the-fly* by the sample confidence at each location, *i.e.*, the more confident locations are more likely to be selected at each step [4, 29]. In contrast, MAR adopts a *fully randomized* order, and its temperature sampling is applied to each token. Table 5 compares this difference in controlled settings. The first line is our MAR implementation but using MAGE's on-the-fly ordering strategy, which has similar results as the simpler random order counterpart. Fully randomized ordering can make the training and inference process consistent regarding the distribution of orders; it also allows us to adopt token-wise temperature sampling in a way similar to autoregressive language models (*e.g.*, GPT [38, 39, 3]).

|  | order | loss | FID↓ | IS↑ |
|---|---|---|---|---|
| MAR, our impl. | on-the-fly | CrossEnt | 8.72 | 145.6 |
| MAR, our impl. | random | CrossEnt | 8.79 | 146.1 |
| MAR, our impl. | random | Diff Loss | 3.50 | 201.4 |

Table 5: To compare conceptually with MAGE, we run MAR's inference using the MAGE strategy of determining the order on the fly by confidence sampling across the spatial domain. These entries are all based on the tokenizers provided by the LDM codebase [42].

Table 6: System-level comparison on **ImageNet 512×512** conditional generation. MAR's CFG scale is set to 4.0; other settings follow the MAR-L configuration described in Table 4.

| | | w/o CFG | | w/ CFG | |
|---|---|---|---|---|---|
| | #params | FID↓ | IS↑ | FID↓ | IS↑ |
| *pixel-based* | | | | | |
| ADM [10] | 554M | 23.24 | 58.1 | 7.72 | 172.7 |
| VDM++ [26] | 2B | 2.99 | **232.2** | 2.65 | 278.1 |
| *vector-quantized tokens* | | | | | |
| MaskGIT [4] | 227M | 7.32 | 156.0 | - | - |
| MAGVIT-v2 [55] | 307M | 3.07 | 213.1 | 1.91 | **324.3** |
| *continuous-valued tokens* | | | | | |
| U-ViT-H/2-G [2] | 501M | - | - | 4.05 | 263.8 |
| DiT-XL/2 [37] | 675M | 12.03 | 105.3 | 3.04 | 240.8 |
| DiffiT [19] | - | - | - | 2.67 | 252.1 |
| GIVT [48] | 304M | 8.35 | - | - | - |
| EDM2-XXL [25] | 1.5B | **1.91** | - | 1.81 | - |
| MAR-L, Diff Loss | 481M | 2.74 | 205.2 | **1.73** | 279.9 |

# D    Additional Comparisons

## D.1    Autoregressive Image Generation in Pixel Space

Our MAR+DiffLoss approach can also be directly applied to model the RGB pixel space without the need for an image tokenizer. To demonstrate this, we conducted an experiment on ImageNet 64×64, grouping every 4×4 pixels into a single token for the Diffusion Loss to model. A MAR-L+DiffLoss model trained for 400 epochs achieved an FID of **2.93**, demonstrating the potential to eliminate the need for tokenizers in autoregressive image generation. However, as commonly observed in the diffusion model literature, directly modeling the pixel space is significantly more computationally expensive than using a tokenizer. For MAR+DiffLoss, directly modeling pixels at higher resolutions might require either a much longer sequence length for the autoregressive transformer or a substantially larger network for the Diffusion Loss to handle larger patches. We leave this exploration for future work.

## D.2    ImageNet 512×512

Following previous works, we also report results on ImageNet at a resolution of 512×512, compared with leading systems (Table 6). For simplicity, we use the KL-16 tokenizer, which gives a sequence length of 32×32 on a 512×512 image. Other settings follow the MAR-L configuration described in Table 4. Our method achieves an FID of 2.74 without CFG and 1.73 with CFG. Our results are competitive with those of previous systems. Due to limited resources, we have not trained the larger MAR-H on ImageNet 512×512, which is expected to have better results.

## D.3    L2 Loss vs. Diff Loss

A naïve baseline for continuous-valued tokens is to compute the Mean Squared Error (MSE, *i.e.*, L2) loss directly between the predictions and the target tokens. In the case of a raster-order AR model, using the L2 loss introduces no randomness and thus cannot generate diverse samples. In the case of the MAR models with the L2 loss, the only randomness is the sequence order; the prediction at a location is deterministic for any given order. In our experiment, we have trained an MAR model with the L2 loss, which as expected leads to a disastrous FID score (>100).

