# OpenReview forum: "Autoregressive Image Generation without Vector Quantization"
_NeurIPS.cc/2024/Conference — NeurIPS 2024 spotlight_

### Official Review · Reviewer_oAwj · 2024-07-13

**Soundness:** 4
**Presentation:** 3
**Contribution:** 3
**Rating:** 7
**Confidence:** 4

**Summary:**

This paper deals with the task of autoregressive image generation using continuous tokenizers. AR image generation has primarily focused on using discrete tokens, and training discrete tokenizers are quite hard. In this paper, the authors train AR models on continuous tokenizers. The idea is to predict continuous valued latent vectors z using a transformer, and modeling the token distribution using a diffusion model conditioned on the latents z. The use of shallow networks for the diffusion helps the models enjoy the fast inference benefit of AR models. Extensive experiments show that diffusion loss can improve the performance of AR models on Imagenet benchmark using various tokenizers.

**Strengths:**

- Use of diffusion loss for AR models is very interesting. The idea is simple, clean and neat. The paper is well written.
- Experiments are extensively performed on different tokenizers.
- Performance gains are solid on different settings.

**Weaknesses:**

- One weakness with this approach is the slow inference speed of diffusion. The authors suggest that a small MLP is sufficient to model the distribution, so it is fast. But looking at the table 3, increasing the model size improves the performance. So, I wonder if in large-scale text-to-image benchmarks, it might be desirable to use large MLPs. If that is the case, the speed could be slower.
- Another weakness is performing experiments only on Imagenet benchmark. As noted by authors, experiments on Imagenet always have noticeable artifacts. It is understandable that some academic labs might not have resources to run experiments on large-scale benchmarks, so I am not going to penalize for this. But, it would have been really nice to see experiments on large-scale text-to-image benchmarks.

**Questions:**

- Can the authors comment on the design choice of using diffusion loss to model P(x|z)? What is the reason for using diffusion? Did you think about other ways of modeling the distribution. One approach which I can think of is to simply represent P(x|z) as a gaussian similar to VAE. Can such simple distributions suffice?

---

> ### Author Rebuttal · Authors · 2024-08-06
>
> We thank the reviewer for appreciating our clean and neat approach as well as our solid empirical results. Here, we address the weaknesses (***W***) and questions (***Q***) raised by the reviewer.
>
> ***Inference speed (W1)***
>
> We agree that DiffLoss can introduce overhead to the inference process, and there is always a trade-off between a larger MLP with more capacity and increased computation overhead. In our ImageNet experiment, a small MLP with a depth of 3 and a width of 1024 achieved favorable performance (1.97 FID) with marginal computational overhead during inference. For large text-to-image datasets, we assume that increasing the capacity of the autoregressive transformer will be more important than increasing the capacity of the MLP in Diffusion Loss. This is because the role of Diffusion Loss is to model a single token with a few dimensions (e.g., 16), which remains a relatively easy task even on large-scale datasets.
>
> ***More experiments (W2)***
>
> We have provided an additional experiment on ImageNet 512x512 in the general response to validate our model’s ability to generalize across different resolutions. Due to resource and time constraints, we are unable to conduct extensive experiments on large-scale text-to-image datasets. However, similar to reviewer xuXX, we believe that the simplicity and standard implementation of the proposed Diffusion Loss makes it easy to be generalized to other datasets and tasks.
>
> ***Other choices to model $p(x|z)$ (Q1)***
>
> This is an excellent question. Conceptually, $p(x|z)$ can be modeled by any conditional generative model and is not limited to diffusion models. We chose diffusion models due to their simplicity and superior performance, as demonstrated in recent literature. We believe that other generative models, such as VAE, GAN, and consistency models, also have the potential to model $p(x|z)$. Exploring these alternatives would be a very interesting future direction.

---

### Official Review · Reviewer_a8ib · 2024-07-14

**Soundness:** 2
**Presentation:** 2
**Contribution:** 2
**Rating:** 3
**Confidence:** 4

**Summary:**

This paper proposes an autoregressive modeling method without the use of vector quantization tokens. By using a diffusion procedure to model the next-token probability, it is able to apply autoregressive models in a continuous-valued space. To model the probability of one token $x_i$ given the condition $X_{<i}$, the output feature of $X_{<i}$ is treated as a condition in the diffusion process of $x_i$. Afterwards,  the authors propose to unify masked modeling and autoregressive modeling through a semi-autoregressive way. Comprehensive experiments show the efficacy of the proposed method.

**Strengths:**

* The idea of integrating the diffusion model and autoregressive model is great.
* The experiments are comprehensive and they seem to support the conclusion that the proposed method outperform the previous SOTA.

**Weaknesses:**

Although the experimental results seem great, there exist several non-negligible weaknesses in this paper:
* The motivation of this paper seems weird. The authors claim that using vector quantization is not a necessity for autoregressive modeling. However, the authors neither provide sufficient examples supporting "Conventional wisdom holds that autoregressive models for image generation are typically accompanied by vector-quantized tokens." as stated in the abstract, nor demonstrate the unnecessity of vector quantization. Since previous works found it work well, why should we discard it?
* The proposed method does not match the motivation. The authors first claim that vector quantization is not necessary in autoregressive modeling. Then the authors propose to fomulate the objective with a diffusion process, which is also complicated and hard to converge. So it seems to me that the authors just replace a complicated method with another complicated method.
* The authors claim that they propose the unification of autoregressive and masked generative models by predicting group by group instead of token by token and in raster order instead of in fixed order. However, these ideas have already been proposed in previous works. The pioneer work should be [1], which first proposed predicting tokens group by group. There are also works on masked image modeling exploring combining masked image modeling and autoregressive modeling [2, 3]. Specifically, [3] explored exactly the same thing in Section 3 as the authors do in the paper. So it seems that this unification is not a novel idea.
* The proposed method still needs a discrete tokenizer, as shown in the experiment section. Are there any experiment results that do not rely on the discrete tokenizer?

[1] Semi-Autoregressive Neural Machine Translation, EMNLP 2018.

[2] Self-supervision through Random Segments with Autoregressive Coding (RandSAC), ICLR 2023.

[3] Bridging Autoregressive and Masked Modeling for Enhanced Visual Representation Learning, https://openreview.net/forum?id=KUz8QXAgFV

**Questions:**

Besides the above questions, there are one question concerning the implementation of the method:
* How do the authors solve the problem of position ambiguity in predicting several tokens at a time in MAR pretraining? To be specific, if the standard ViT structure is used as stated in the paper, when predicting a group of tokens, how does the model know which position is being predicted? This problem has also been stated in [3, 4].

[3] Bridging Autoregressive and Masked Modeling for Enhanced Visual Representation Learning, https://openreview.net/forum?id=KUz8QXAgFV

[4] XLNet: Generalized Autoregressive Pretraining for Language Understanding, https://arxiv.org/abs/1906.08237

**Limitations:**

The authors have stated the limitations in Appendix A.

---

> ### Author Rebuttal · Authors · 2024-08-06
>
> We thank the reviewer for appreciating our approach and the comprehensive experimental results. It seems there is a key misunderstanding about our paper that “the proposed method still needs a discrete tokenizer”. In fact, ***ALL*** experimental results in our paper using Diffusion Loss are conducted with ***continuous KL-based tokenizers***, except for the first row in Table 2, where we show that Diffusion Loss can also work well with a discrete VQ-based tokenizer. We believe this misunderstanding may be the primary reason the reviewer questions our motivation and method. We hope the reviewer will reconsider the rating in light of this clarification. Below, we address the specific weaknesses (***W***) and questions (***Q***) raised.
>
> ***Motivation (W1)***
>
> The reviewer questions the motivation of the paper, stating that "the authors neither provide sufficient examples supporting 'conventional wisdom holds that autoregressive models for image generation are typically accompanied by vector-quantized tokens,' nor demonstrate the unnecessity of vector quantization.” We respectfully disagree with these two statements. As stated in L21-25 and 68-70, almost all recent works on autoregressive image generation rely on vector-quantized tokens, starting from VQGAN [1] and DALL-E [2] to MaskGIT [3] and MAGE [4], and more recently, VAR [5] and TiTok [6]. As a side proof, Reviewer xuXX mentioned, "I always believed that sticking to discrete tokens was necessary to train AR models," and Reviewer V3y3 noted, "the paper introduces a novel approach of using continuous-valued tokens in autoregressive models for image generation, challenging the conventional wisdom of discrete vector-quantized tokens." These comments strongly support the notion that "conventional wisdom holds that autoregressive models for image generation are typically accompanied by vector-quantized tokens."
>
> Moreover, as mentioned earlier, all major experimental results in our paper using Diffusion Loss DO NOT use VQ-based tokenizers. In Table 1, we show that Diffusion Loss, when used with a continuous KL-16 tokenizer, outperforms its cross-entropy counterpart with a discrete VQ-16 tokenizer, consistently observed across all variants of AR and MAR. This result clearly demonstrates the unnecessity of vector quantization and the superior advantage of using Diffusion Loss to model continuous-valued tokens.
>
> ***Complication of Diffusion Loss (W2)***
>
> We respectfully disagree with the statement that “the diffusion process is complicated and hard to converge.” The intuition behind the diffusion model is quite simple: adding and removing noise. The mathematical formulation is also straightforward, involving a standard SDE equation. Many generative modeling works in recent years have empirically demonstrated that diffusion models are not hard to train and converge, and they have been successfully applied to various domains and problems.
>
> We also respectfully disagree with the reviewer’s statement that “the authors just replace a complicated method with another complicated method.” We use a simple MLP controlled by only two parameters, depth and width, as the denoising network. We employ a very standard diffusion process from iDDPM [7], which is widely used by common diffusion models such as ADM [7] and DiT [8]. The simplicity of Diffusion Loss is also highly appreciated by other reviewers. Reviewer xuXX states that “the architecture does not contain any specialized components or complex diffusion loss.” Reviewer oAwj notes that “the idea is simple, clean, and neat.” Thus, we believe that the proposed Diffusion Loss is a straightforward yet highly effective module that enables autoregressive image generation on continuous-valued tokens, achieving superior performance compared to traditional autoregressive models on discrete-valued tokens.
>
> ***Unification of autoregressive and masked generative models (W3)***
>
> First, we want to clarify that the unification of autoregressive (AR) and masked generative (MAR) models is not intended as a novel technical contribution, as both approaches have been thoroughly explored in the literature [1, 3, 4, 9]. Instead, as stated in L47-51, the unification aims to broaden the scope where DiffLoss can be applied and can be helpful.
>
> Second, the two works [10, 11] mentioned by the reviewer focus on self-supervised representation learning rather than generative models. It's important to note that within the visual generative modeling community, masked generative models are not commonly recognized as autoregressive models. In fact, many of these models are referred to as “non-autoregressive models” in the literature [3, 9]. One goal of this paper is to show that these models still possess the autoregressive characteristic of “predicting next tokens based on known ones,” allowing them to seamlessly use Diffusion Loss. As noted by Reviewer V3y3, “the unification of autoregressive and masked generative models under a generalized framework is a great contribution to the field.”
>
> ***Position ambiguity (Q1)***
>
> Similar to MAE [12], we add a learnable positional embedding to each masked token to indicate its position in the sequence.
>
> [1] Taming Transformers for High-Resolution Image Synthesis
>
> [2] Zero-Shot Text-to-Image Generation
>
> [3] MaskGIT: Masked Generative Image Transformer
>
> [4] MAGE: MAsked Generative Encoder to Unify Representation Learning and Image Synthesis
>
> [5] Scalable Image Generation via Next-Scale Prediction
>
> [6] An Image is Worth 32 Tokens for Reconstruction and Generation
>
> [7] Diffusion Models Beat GANs on Image Synthesis
>
> [8] Scalable Diffusion Models with Transformers
>
> [9] Muse: Text-To-Image Generation via Masked Generative Transformers
>
> [10] Bridging Autoregressive and Masked Modeling for Enhanced Visual Representation Learning
>
> [11] Self-supervision through Random Segments with Autoregressive Coding (RandSAC)
>
> [12] Masked Autoencoders Are Scalable Vision Learners

---

### Official Review · Reviewer_mV98 · 2024-07-21

**Soundness:** 3
**Presentation:** 4
**Contribution:** 3
**Rating:** 8
**Confidence:** 4

**Summary:**

The paper introduces an autoregressive image generation method without vector quantisation for visual. The authors observe that while discrete-valued spaces facilitate representing categorical distributions, they are not necessary for autoregressive modeling. They propose modeling the *per-token* probability distribution using a diffusion process, allowing the application of autoregressive models in a continuous-valued space. This approach eliminates the need for discrete-valued tokenizers and has been experimentally validated across standard autoregressive models and generalized masked autoregressive variants.

**Strengths:**

This work is articulated clearly, with well-founded theoretical support for its motivation. The proposed method effectively addresses the existing problems and offers significant inspiration for future research:

* Clear Motivations: Under the AR, there is no intrinsic connection between the method of next token prediction and quantisation. Using discrete distributions, e.g., categorical or multinomial distributions, for image modeling is also counterintuitive.


* Original Contributions: The work introduced the Diffusion loss with various types of AR to model the distribution of continuous tokens, thereby eliminating the need for discretisation. Although some existing literatures have used diffusion [1] or continuous token modeling [2], this work is the first to combine diffusion with AR for generative tasks with solid experiment results


* Significant Benefits: This work takes a significant step forward in advancing continuous modeling. I personally believe image quantisation is a simplification and compromise to facilitate AR-based generative models, as modeling discrete distributions is much easier. However, quantisation inevitably leads to information loss. The proposed solution effectively mitigates this potential loss. Additionally, continuous representation is more suitable for modeling objective phenomena governed by physical laws, such as images or videos.


[1] Li, Yazhe, Jorg Bornschein, and Ting Chen. "Denoising Autoregressive Representation Learning." arXiv preprint arXiv:2403.05196 (2024).
[2] Tschannen, Michael, Cian Eastwood, and Fabian Mentzer. "GIVT: Generative infinite-vocabulary transformers." arXiv preprint arXiv:2312.02116 (2023).

**Weaknesses:**

### Concerns
Overall, this work is good, but the theoretical analysis is somewhat insufficient.

One concern is that, although quantisation is avoided during generation, the entire process still relies on an encoder/decoder. Can we establish theoretical assumptions to prove that modeling with diffusion loss has lower information loss, or that its upper bound on loss is lower than the version with quantization (although Table 2 has experimentally validated this assumption)?

Another concern is after replaced Cross entropy, can we obtain an explicit prior distribution or posterior for the whole AR system? Sec 3.1 discussed about CE in traditional AR system is for Cat distribution, but I feel I cannot find similar discussions for Diffusion in the following work.

The next question is can we model the pixel space directly instead an encoder? Is this limited by the computation cost?


### Open Questions

An open question:  quantisation is not necessary any more for AR, but is the AR still necessary with diffusion loss? Seems image tokens (whether quantised or not) differ significantly from language attributes.  The prior distribution of image tokens might be uniform thus current AR might still face the same drawbacks?

Another open question is that can we include a decoder in end-to-end training and escape the limitations of the tokenizer?


### Suggestions

For readers, some key links appear to be missing in the reasoning chain, e.g.:

Why we use Diffusion Loss with noisy input instead of clean image with MSE for continuous modelling?  I think one potential explanation is that the objective of MSE is actually minimizing the divergence between Gaussian and token and it differs significantly from the empirical token distribution.

**Questions:**

Please see weaknesses. My Open Questions and Suggestions will not affect my final rating.

**Limitations:**

The author has stated limitations and social impact.

---

> ### Author Rebuttal · Authors · 2024-08-06
>
> We thank the reviewer for the appreciation of the clear motivations, original contributions, and significant benefits of our work. Here we address the weaknesses (***W***) and questions (***Q***)  raised by the reviewer. We are also happy to discuss further if you have any additional questions or confusion.
>
> ***Analysis of Information Loss (W1)***
>
> As pointed out by the reviewer, our experiment results in Table 2 and the attached PDF in the general response demonstrate that the reconstruction quality of VQ-based tokenizers is significantly worse than that of KL-based tokenizers. We believe this is because VQ-based tokenizers experience much more information loss due to their higher compression ratio.
>
> Here, we provide an informal reasoning about the information loss by analyzing the compression ratio of VQ-based and KL-based tokenizers. For example, a VQ-16 tokenizer with a codebook size of 1024 tokenizes a 256x256 image into a 16x16 sequence of discrete indices, each of which can be represented by 10 bits. In contrast, a KL-16 tokenizer encodes the same image into a 16x16x16 sequence of float numbers. Thus, the compression ratio of the VQ-16 tokenizer is much higher than that of the KL-16 tokenizer, resulting in significantly more information loss. Therefore, the lower bound of information loss by modeling KL-16 continuous tokens using Diffusion Loss is lower compared to modeling VQ-16 discrete token indices using cross-entropy loss.
>
> We also believe a formal analysis could further strengthen the theoretical foundation of Diffusion Loss. However, due to the significant efforts required, we leave this for future work.
>
> ***Explicit prior or posterior for the whole AR system (W2)***
>
> One advantage of using a categorical distribution is that it can explicitly compute the probability of each sampled discrete token, which can then be used to compute the posterior of the entire AR system. While diffusion models are typically implicit generative models, [1] has demonstrated that it is possible to compute the exact likelihood of each continuous token using a probability flow ODE. This could potentially enable computing the posterior of the AR+DiffLoss system, offering an intriguing direction for future exploration.
>
> [1] Score-Based Generative Modeling through Stochastic Differential Equations
>
> ***Directly modeling the pixel space (W3)***
>
> This is an excellent question. Diffusion Loss is independent of the tokenizer and can be directly applied in the pixel space. Given the limited time for the rebuttal, we conduct a preliminary experiment on ImageNet 64x64, where we group every 4x4 pixels into one token for Diffusion Loss to model. A MAR-L+DiffLoss model trained for 400 epochs achieved an FID of 2.93, demonstrating the potential to eliminate the need for tokenizers in autoregressive image generation with diffusion loss. However, as commonly observed in the literature on diffusion models, directly modeling the pixel space is much more computationally expensive than using a tokenizer [2]. For MAR+DiffLoss, directly modeling pixels at a higher resolution might require either a much longer sequence length for the autoregressive transformer or a much larger network for Diffusion Loss to handle larger patches.
>
> [2] High-Resolution Image Synthesis with Latent Diffusion Models
>
> ***Is AR still necessary with diffusion loss (Q1)***
>
> We believe that one major advantage of AR/MAR models is their ability to decompose a complex joint probability distribution $p(x_1, \cdots, x_n)$ into the product of multiple simpler conditional distributions $\prod^n_{i=1}  p(x^i~ |~ x^1, ..., x^{i-1})$. Therefore, we believe AR/MAR models will continue to play an important role in modeling complex data distributions, such as high-resolution images or videos. Our paper aims to pave the way for this by eliminating the need for quantization and enabling the autoregressvie modeling of continuous distributions.
>
> ***End-to-end training of decoder (Q2)***
>
> We thank the reviewer for pointing out this possibility. Since Diffusion Loss eliminates the need for vector quantization, the reconstruction loss on pixels can be directly back-propagated to the continuous token output by the AR+DiffLoss part. This makes it possible to train the entire framework in an end-to-end manner. We believe exploring this possibility would be a very interesting future direction.
>
> ***Direct regression using MSE loss (Suggestion)***
>
> We thank the reviewer for the suggestion. A naive baseline to model continuous-valued tokens is to compute the Mean Squared Error (MSE, i.e., L2) loss directly between the predictions and the target tokens. In the case of a raster-order AR model, using the L2 loss introduces no randomness and thus cannot generate diverse samples. In the case of MAR models with L2 loss, the only randomness is the sequence order; the prediction at a location is deterministic for a given order. In our experiment, we trained an MAR model with L2 loss, which, as expected, led to a disastrous FID score ($>$100). We will include this in the revision.

---

### Official Review · Reviewer_xuXX · 2024-07-24

**Soundness:** 3
**Presentation:** 3
**Contribution:** 3
**Rating:** 8
**Confidence:** 5

**Summary:**

This work challenges the conventional belief that autoregressive (AR) models implemented by a Transformer are best suited for modeling discrete sequences. Instead, it proposes modeling the per-token probability distribution using a diffusion procedure, enabling the application of AR models in a continuous-valued space. By defining a Diffusion Loss function instead of using categorical cross-entropy loss, this approach eliminates the need for discrete-valued tokenizers. Consequently, this framework can directly model unquantized continuous-valued tokens, avoiding information loss and degraded reconstruction performance associated with discretized tokens. Therefore, it is expected to provide better generation quality with the same compression ratio compared to AR models with discrete sequences. Experiments on class-conditional image generation demonstrate that this framework achieves strong results while benefiting from the speed advantage of sequence modeling.

**Strengths:**

#1. Presentation and story-telling. In my personal experience, while trying to improve VQ-GAN in various ways, I always believed that sticking to discrete tokens was necessary to train AR models. This work has a very compelling motivation to challenge this conventional belief, which many researchers and developers in this field presumably hold. I appreciate how the manuscript effectively describes the motivation and explains why quantization is not essential for training AR models.

#2. This work proposes a novel and interesting idea. This generalizable framework has full potential and could serve as a fundamental building block in multi-modal LMM, effectively handling multi-modal data naturally, rather than relying on adaptor-based approaches. Previous attempts required continuous signals to be vector-quantized before training an autoregressive model with a Transformer architecture. However, this framework overcomes those limitations, advancing the field in various ways.

#3. Simplicity of implementation. The architecture does not contain any specialized components or complex diffusion loss, which increases my confidence in being able to reproduce the results. Furthermore, all important experimental details are thoroughly described in the manuscript.

**Weaknesses:**

The main weakness of this work is the limited scope of experiments, as the manuscript only presents ImageNet 256x256 class-conditional experiments. However, the authors acknowledge this limitation in the Appendix. Additionally, I am confident that the benefits of diffusion loss validated by ImageNet will be generalizable to other datasets and tasks, including text-to-image generation. Therefore, this weakness does not outweigh the strengths of this work.

**Questions:**

#1. In the manuscript, the authors mention that adding 64 CLS tokens at the start of the sequence helps improve the stability and capacity of our encoding when training masked generative models with bidirectional attention. Could you elaborate on why this improves stability? Is it because it sets the minimum sequence length to 64?

#2. "Vector-quantized tokenizers are difficult to train and are sensitive to gradient approximation strategies." Can we say this is difficult to train? While dealing with a large codebook size may require special treatment, such as restarting dead codes, training VQ-VAE or VQ-GAN with a moderate codebook size may not be difficult once one is familiar with the framework. This suggests that there is no inherent art to it.

#3. Does "16 denotes the tokenizer strides in line 219" refer to the downsampling factor?

#4. Why is the gap between CrossEnt and Diff Loss in Table 1 smaller for the AR model compared to the gap for the MAR model?

#5. Experiments involving tokenizers with mismatched strides are intriguing. However, I didn't understand why we are considering using these tokenizers, given that KL-8 does not outperform KL-16 despite having the same training and inference costs.

#6. Could you elaborate on why AR with random order improves fidelity compared to AR with a raster scan order?

**Limitations:**

The limitations of this work are thoroughly discussed in the Appendix. No societal negative impacts have been observed.

---

> ### Author Rebuttal · Authors · 2024-08-06
>
> We thank the reviewer for the appreciation of our motivation and proposed method. Here we address the weaknesses (***W***) and questions (***Q***) raised by the reviewer.
>
> ***Scope of experiments (W1)***
>
> We have provided an additional experiment on ImageNet 512x512 in the general response to validate our model’s ability to generalize across different resolutions. Due to resource and time constraints, we are unable to conduct extensive experiments on large-scale text-to-image datasets. However, we agree with the reviewer that the simplicity and standard implementation of the proposed Diffusion Loss make it easy to be generalized to other datasets and tasks, including text-to-image generation.
>
> ***Adding CLS tokens (Q1)***
>
> If no additional padding is used (i.e., only one [CLS] token), the encoder's input sequence can become very short at high masking ratios, sometimes as short as one token when the masking ratio is 100%. This can make the training unstable and hurt the generation performance due to the limited amount of computation spent in the encoder when the masking ratio is high. To address this, we pad the encoder sequence with 64 [CLS] tokens at the beginning, ensuring a minimum sequence length of 64. This padding stabilizes training and enhances the encoder's computation, leading to improved generation performance.
>
> ***The difficulty in training VQ-based tokenizers (Q2)***
>
> We apologize for the confusion. By “difficult to train,” we mean that VQ-based tokenizers typically require much more specialized techniques compared to standard KL-based tokenizers to achieve good reconstruction performance. These techniques include replacing dead codes, L2 normalization on the encoded latent variables, logit-Laplace loss, affine reparameterization of the code vectors, synchronized and alternating training, etc [1, 2]. This also introduces many additional hyper-parameters to tune. With similar specialized techniques and hyper-parameter tuning, the reconstruction quality of VQ-based tokenizers can lag significantly behind their continuous KL-based counterparts, as shown in the attached PDF in the general response. We will clarify this point in the revision.
>
> [1] Straightening Out the Straight-Through Estimator: Overcoming Optimization Challenges in Vector Quantized Networks.
>
> [2] Vector-quantized Image Modeling with Improved VQGAN
>
> ***Tokenizer stride (Q3)***
>
> Yes, tokenizer stride means the downsampling factor of the tokenizer.
>
> ***The gap between CrossEnt and Diff Loss is smaller for the AR model than the MAR model (Q4)***
>
> We speculate that the generative power of the AR model is limited by its raster order and causal attention mechanism, which are not as well-suited for image generation as random order and bidirectional attention. Thus, the bottleneck for the AR model may lie in the generative capacity of the backbone transformer architecture rather than the nature of the data distribution, whether continuous or discrete. In contrast, the MAR model has sufficient capacity to effectively model the data distribution, with the primary bottleneck being the information loss caused by the VQ tokenizer. This might explain why Diffusion Loss sees larger gains in MAR models than in AR models. A definitive answer to this question requires further extensive experiments, which we leave as future work.
>
> ***Tokenizers with mismatched strides (Q5)***
>
> The purpose of the experiments with KL-16 and KL-8 is to demonstrate the flexibility provided by Diffusion Loss, which decouples the downsampling factor of the tokenizer from the sequence length of the autoregressive transformer. This flexibility eliminates the need to retrain the tokenizer for each sequence length. More importantly, similar to what is seen in DiT, it allows for a customizable choice of the transformer's sequence length, enabling a tailored trade-off between computational costs and performance.
>
> ***Random order vs. raster order (Q6)***
>
> We note that random order does not necessarily improve fidelity when comparing the performance of raster-order and random-order AR models with classifier-free guidance. While the random-order AR models achieve a better FID, they have a worse IS. IS primarily measures fidelity, whereas FID measures both fidelity and diversity. The improvement in FID is likely due to the global randomness introduced by the random order, which enhances the diversity of the generated images.

---

### Official Review · Reviewer_V3y3 · 2024-07-29

**Soundness:** 4
**Presentation:** 4
**Contribution:** 3
**Rating:** 7
**Confidence:** 5

**Summary:**

This paper introduces a novel approach to image generation using autoregressive models with continuous-valued tokens, challenging the conventional use of discrete vector-quantized tokens. The authors propose a "Diffusion Loss" function that models per-token probability distributions using a diffusion process, eliminating the need for vector quantization. They demonstrate the effectiveness of this approach across various autoregressive models, including standard and masked variants. The paper also unifies autoregressive and masked generative models under a generalized framework, showing that bidirectional attention can perform autoregression. The authors implement their method using a small denoising MLP and evaluate it on ImageNet, achieving state-of-the-art results for token-based image generation. They demonstrate the flexibility of Diffusion Loss with various tokenizers and explore its properties, such as temperature control and sampling steps. The approach shows favorable speed-accuracy trade-offs compared to existing methods like Diffusion Transformers. Overall, this work opens up new possibilities for autoregressive modeling in continuous-valued domains, potentially impacting various applications beyond image generation.

**Strengths:**

- The paper introduces a novel approach of using continuous-valued tokens in autoregressive models for image generation, challenging the conventional wisdom of discrete vector-quantized tokens. The proposed Diffusion Loss is an innovative way to model per-token probability distributions, bridging the gap between autoregressive and diffusion models. The unification of autoregressive and masked generative models under a generalized framework is a great contribution to the field.
- The empirical results are robust and comprehensive, demonstrating consistent improvements across various model variants and tokenizers.
- The ablation studies and analyses of different components (e.g., denoising MLP, sampling steps, temperature) are thorough and insightful.
- The speed-accuracy trade-off analysis provides a practical perspective on the method's performance.
- The paper is well-structured and logically organized, guiding the reader through the concept, implementation, and results. The use of figures (especially Figures 2 and 3) effectively illustrates complex concepts like bidirectional attention for autoregression and the generalized autoregressive framework.

**Weaknesses:**

- While the paper demonstrates good speed-accuracy tradeoffs, I'm concerned about the overall computational complexity, especially during training. The diffusion process adds significant overhead, and training for 400 epochs seems quite intensive. How does the training time and compute requirements compare to other state-of-the-art methods?
- While the paper explores different model sizes, a more systematic study of how performance scales with model size (similar to studies in language models) could provide valuable insights into the method's potential for further improvements. Usually diffusion models can be more parameter efficient compared to autoregressive models, I'm curious to see if it's still the case for the autoregressive diffusion modeling which is a hybrid between the two.
- For the interpretability of the continuous-valued tokens or the learned representations. An analysis of what these tokens capture compared to discrete tokens could provide valuable insights into why the method works so well.

**Questions:**

- Can authors elaborate more on the rationale behind the idea of using small NLP for denoising conditioning on latent code produced by the transformer? Like comparing to diffusion training without conditioning on latent $z$ ?
- Have you experimented with the same model architecture but applied directly to pixels or image patches, rather than using image tokenizers? How significant is the use of more contextualized tokenized codes compared to working with pixels directly?
- Is there a theoretical limit to how much continuous-valued tokens can improve performance over discrete ones? Are there any situations where discrete tokens might actually be better? The paper seems to assume continuous tokens are always superior, but is there any proof of this, or are there any counterexamples where this might not be true?
- Regarding artifact handling, do you think the issues are more related to the diffusion modeling process, or are they inherent limitations of the image tokenizers themselves?

**Limitations:**

Yes.

---

> ### Author Rebuttal · Authors · 2024-08-06
>
> We thank the reviewer for appreciating our novel approach, significant contribution, and comprehensive empirical results. Here, we address the weaknesses (***W***) and questions (***Q***) raised by the reviewer.
>
> ***Computational complexity (W1)***
>
> |Methods| Epochs |Training Costs (days)|FID w/o CFG|FID w/ CFG
> |-|-|-|-|-|
> |LDM-4|165|3.93|10.56|3.60|
> |DiT-XL/2|1400|16.1|9.62 |2.27|
> |MAR+CrossEnt|400|2.25|8.79|3.69|
> |MAR+DiffLoss|400|2.57|**3.50**|**1.98**|
>
> The table above compares the training times of different methods on our cluster of 128 V100 GPUs. While the diffusion loss introduces some training overhead compared to cross-entropy, this overhead is relatively marginal to the computational cost of the backbone autoregressive transformer, resulting in a total training cost increase of only 14%. Additionally, the training cost of our model remains significantly lower than that of representative diffusion-based models such as DiT-XL/2 and LDM-4.
>
> ***Performance vs. #Parameters (W2)***
>
> In Table 4 of the paper, we evaluated MAR's performance with different numbers of parameters and demonstrated promising scaling signals up to 1 billion parameters. Due to resource constraints, we are unable to conduct an extensive systematic evaluation of the scaling behavior and will leave this for future work. MAR-L+DiffLoss achieves an FID of 1.78 with 479M parameters, while a representative diffusion model, DiT-XL/2, achieves an FID of 2.27 with 675M parameters. This demonstrates that MAR+DiffLoss can be more parameter efficient than common diffusion models. This is likely due to the performance improvement brought by DiffLoss, as shown in Table 1 of the paper.
>
> ***Continuous tokens vs. discrete tokens (W3, Q3)***
>
> We provide visualizations of the reconstruction performance using VQ-based and KL-based tokenizers in the attached PDF in the general response. The results clearly demonstrate that the reconstruction quality of VQ-based tokenizers is significantly worse than that of KL-based tokenizers. The KL-16 tokenizer can reconstruct far more details from the original image. For example, the face of the Mona Lisa reconstructed by the KL-16 tokenizer is much better than that reconstructed by the VQ-16 tokenizer.
>
> This is because VQ-based tokenizers experience much more information loss due to the higher compression ratio of the quantized tokens: for example, a VQ-16 tokenizer with a codebook size of 1024 tokenizes a 256x256 image into a 16x16 sequence of discrete indices, each of which can be represented by 10 bits. In contrast, a KL-16 tokenizer encodes the same image into a 16x16x16 sequence of float numbers. Thus, the compression ratio of the VQ-16 tokenizer is much higher than that of the KL-16 tokenizer, resulting in significantly more information loss. Therefore, the lower bound of information loss by modeling KL-16 continuous tokens using Diffusion Loss is lower compared to modeling VQ-16 discrete token indices using cross-entropy loss. We believe this may explain why Diffusion Loss on continuous-valued tokens can significantly outperform cross-entropy on discrete tokens.
>
> On the other hand, one possible advantage of discrete-valued tokens is their ability to explicitly compute the probability of each sampled discrete token, which is important for certain sampling methods such as beam search. Although prior works have shown that diffusion models can also compute the explicit probability density on the continuous distribution using a probability-flow ODE [1], such computation is relatively less straightforward.
>
> ***The rationale behind using a small MLP conditioning on latent code (Q1)***
>
> We use a small MLP conditioned on the latent code $z$ to model the probability distribution $p(x|z)$ through a diffusion process. Since the dimension of $x$ is small (16 for KL-16), the MLP can be small yet still accurately model $p(x|z)$. Such conditioning is also key to training the autoregressive transformer, as the diffusion loss is backpropagated to the large autoregressive transformer through $z$. Without conditioning on $z$, the MLP cannot effectively model $p(x|z)$, nor can the autoregressive transformer be properly trained.
>
> ***Directly modeling the pixel space (Q2)***
>
> This is an excellent question, as also pointed out by reviewer mV98. Diffusion Loss does not depend on the tokenizer and can thus be applied directly to pixel space. Given the limited time for the rebuttal, we conducted a preliminary experiment on ImageNet 64x64, grouping every 4x4 pixels into one token for Diffusion Loss to model. A MAR-L+DiffLoss model trained for 400 epochs achieved an FID of 2.93, demonstrating the potential to eliminate the need for tokenizers in autoregressive image generation.
> However, as commonly observed in the literature on diffusion models, directly modeling the pixel space is much more computationally expensive than using a tokenizer [2]. For MAR+DiffLoss, directly modeling pixels at a higher resolution might require either a much longer sequence length for the autoregressive transformer or a much larger network for Diffusion Loss to handle larger patches.
>
> ***Artifact (Q4)***
>
> We believe this issue is not primarily due to the KL-based continuous image tokenizer or the diffusion process. Instead, we think it is mostly due to the dataset size. The artifact problem is commonly observed when models are trained only on ImageNet (DiT also exhibits artifacts). Similar tokenizers and diffusion models are widely used in commercial models like Stable Diffusion, which experience far fewer artifacts, likely because they are trained on the much larger LAION-2B dataset. A concrete example is that when trained on LAION-400M, LDM [2] achieves significantly better performance compared to training on ImageNet, even though the model architecture and tokenizer remain almost the same.
>
> [1] Score-Based Generative Modeling through Stochastic Differential Equations
>
> [2] High-Resolution Image Synthesis with Latent Diffusion Models

---

### Author Rebuttal · Authors · 2024-08-06

We thank all reviewers for providing lots of insightful and constructive feedback. We will definitely improve our manuscript accordingly. We are glad to see the commonly recognized strengths highlighted by the reviewers:

1. The presentation of the paper is clear and well-structured (xuXX, mV98, V3y3, oAwj).

2. The paper makes original and significant contributions to the field (xuXX, mV98, V3y3). The introduced idea is great (a8ib), interesting (oAwj), and novel (V3y3, xuXX).

3. The paper provides a clear and compelling motivation to move away from discrete-valued tokenizers and offers a thorough explanation of why quantization is not essential for AR models (xuXX, mV98).

4. The proposed Diffusion Loss is clean, neat (oAwj), and simple to implement (xuXX).

5. The empirical results are solid (mV98, oAwj) and comprehensive (V3y3, a8ib).

We would also like to present additional experimental results on ImageNet 512x512 (Table 1 in the attached PDF), where our MAR-L+DiffLoss model also achieves state-of-the-art performance (1.73 FID with CFG). This demonstrates our model’s ability to generalize and perform well across different resolutions.

In the attached PDF file we further compare the reconstruction performance of VQ-16 and KL-16 tokenizers provided by LDM (Figure 1). The reconstruction performance of KL-16 is clearly better than that of VQ-16, which further highlights the advantage of the proposed DiffLoss which can perform autoregressive modeling on continuous tokens provided by KL-16.

As there are no outstanding common questions, we will address each reviewer’s specific questions in separate responses. We are also happy to continue the discussion if the reviewers have any further questions or concerns.

---

### Decision · Program_Chairs · 2024-09-25

**Decision:**

Accept (spotlight)

**Comment:**

This paper introduces a novel approach to autoregressive image generation by eliminating the need for discrete vector-quantized tokens, using diffusion loss in continuous-valued spaces. Reviewers broadly agree that the work is a strong and impactful contribution. Reviewer V3y3 highlighted the novelty and potential of the approach to bridge autoregressive and diffusion models, providing a unified framework that could extend to various applications beyond image generation. Reviewer xuXX appreciated the simplicity and straightforward implementation of the method, which increases reproducibility without requiring complex components. The empirical evaluations are robust and comprehensive, demonstrating clear performance gains across different models and settings, as noted by V3y3 and mV98. The paper is also well-organized and clearly presents its contributions, making it accessible to a broad audience. Some concerns were raised about computational complexity and the limited scope of experiments mainly on ImageNet. However, these were effectively addressed by the authors, who demonstrated that the additional overhead is manageable and that the method’s potential extends beyond the datasets used. Overall, the paper offers a significant and well-supported contribution to generative modeling with clear potential for future applications, warranting its acceptance.